# Utilizing Extraepitopic Amino Acid Substitutions to Define Changes in the Accessibility of Conformational Epitopes of the *Bacillus cereus* HlyII C-Terminal Domain

**DOI:** 10.3390/ijms242216437

**Published:** 2023-11-17

**Authors:** Natalia V. Rudenko, Alexey S. Nagel, Bogdan S. Melnik, Anna P. Karatovskaya, Olesya S. Vetrova, Anna V. Zamyatina, Zhanna I. Andreeva-Kovalevskaya, Alexander V. Siunov, Mikhail G. Shlyapnikov, Fedor A. Brovko, Alexander S. Solonin

**Affiliations:** 1Pushchino Branch, Shemyakin–Ovchinnikov Institute of Bioorganic Chemistry, Russian Academy of Sciences, 6 Prospekt Nauki, 142290 Pushchino, Moscow Region, Russia; bmelnik@phys.protres.ru (B.S.M.); annakaratovskaya@mail.ru (A.P.K.); olesja.wetrowa1999@gmail.com (O.S.V.); anna.zamjatina@yandex.ru (A.V.Z.); brovko@bibch.ru (F.A.B.); 2G.K. Skryabin Institute of Biochemistry and Physiology of Microorganisms, Russian Academy of Sciences, FRC Pushchino Scientific Centre of Biological Research, Russian Academy of Sciences, 5 Prospekt Nauki, 142290 Pushchino, Moscow Region, Russia; anagell@mail.ru (A.S.N.); hemolysin6@gmail.com (Z.I.A.-K.); av_siunov@rambler.ru (A.V.S.); shlyapnikov@ibpm.pushchino.ru (M.G.S.); solonin.a.s@yandex.ru (A.S.S.); 3Institute of Protein Research, Russian Academy of Sciences, 4 Institutskaya Street, 142290 Pushchino, Moscow Region, Russia

**Keywords:** pore-forming toxin, monoclonal antibodies, epitope, site-directed mutagenesis, enzyme immunoassay, phage display, modeling of three-dimensional structures

## Abstract

Hemolysin II (HlyII)—one of the pathogenic factors of *Bacillus cereus*, a pore-forming β-barrel toxin—possesses a C-terminal extension of 94 amino acid residues, designated as the C-terminal domain of HlyII (HlyIICTD), which plays an important role in the functioning of the toxin. Our previous work described a monoclonal antibody (HlyIIC-20), capable of strain-specific inhibition of hemolysis caused by HlyII, and demonstrated the dependence of the efficiency of hemolysis on the presence of proline at position 324 in HlyII outside the conformational antigenic determinant. In this work, we studied 16 mutant forms of HlyIICTD. Each of the mutations, obtained via multiple site-directed mutagenesis leading to the replacement of amino acid residues lying on the surface of the 3D structure of HlyIICTD, led to a decrease in the interaction of HlyIIC-20 with the mutant form of the protein. Changes in epitope structure confirm the high conformational mobility of HlyIICTD required for the functioning of HlyII. Comparison of the effect of the introduced mutations on the effectiveness of interactions between HlyIICTD and HlyIIC-20 and a control antibody recognizing a non-overlapping epitope enabled the identification of the amino acid residues N339 and K340, included in the conformational antigenic determinant recognized by HlyIIC-20.

## 1. Introduction

*Bacillus cereus* are opportunistic gram-positive spore-forming bacteria [1,2]. One of the key factors of bacterial pathogenicity is hemolytic toxin II (hemolysin II, HlyII), which belongs to the group of β-pore-forming toxins [3,4]. This toxin is secreted by bacteria in the form of water-soluble monomers and oligomerizes in the presence of the plasma membrane of the target cell to form transmembrane pores, leading to a disruption of osmotic pressure inside the cell and ultimately to its destruction, providing bacteria with access to nutrients [5,6]. HlyII of *B. cereus* is capable of lysing erythrocytes [5] and other eukaryotic cells [7,8,9]. It differs from other β-pore-forming toxins due to its presence of a C-terminal extension of 94 amino acid residues, the C-terminal domain of HlyII (HlyIICTD) [10]. Elimination of the C-terminal domain reduces the hemolytic activity toward rabbit erythrocytes eightfold [5]. HlyIICTD has been shown to have the independent ability to orientedly bind to target cell membranes. In the presence of cell and artificial membranes, it is capable of oligomerizing and forming ion channels. Under a prolonged exposure to HlyIICTD lyses representative human cell lines: monocytes, macrophages, and T cells [11].

The structure of HlyIICTD determined via NMR is a folded pseudo-barrel consisting of two α-helices surrounded by five β-sheets [12]. Studying the role of HlyII individual domains can bring us closer to understanding the water-soluble state of this toxin and the stages of its secretion from bacterial cells into the external environment, as well as its pore formation stages. It has been previously assumed that the wild-type HlyIICTD domain exists in two states due to the *cis*-/*trans*-isomerization of proline at position 405 of the full-length HlyII toxin [13], which can lead to a doubling of resonance signals. NMR study of HlyIICTD structure has demonstrated the uniqueness of this domain. No molecules of a similar spatial structure have been found to date [12,13]. Presumably, HlyIICTD forms a separate domain and is close to the core part of the full-length molecule [12]. Due to a problem that could arise in this case, NMR analysis is performed on the mutant form, where the proline residue is replaced by methionine. This mutant form exists only in the *trans* state and provides a model of this domain in the context of the full-length HlyII protein [12,14].

In a previous report [15], we have described that the HlyIIC-20 monoclonal antibody is capable of strain-specific suppression of hemolysis caused by *B. cereus* HlyII. It was shown that the ability of this antibody to suppress hemolysis depends on the presence of Leu or Pro at position 324 in the full-length toxin molecule. This work, using site-directed mutagenesis, which leads to the replacement of amino acid residues located on the surface of the spatial structure of HlyIICTD (6d5z), determined which amino acid residues are included in the HlyIIC-20 epitope.

## 2. Results

### 2.1. Monoclonal Antibodies HlyIIC-20 and HlyIIC-40 Recognize Non-Overlapping Regions on the Surface of the Spatial Structure of HlyIICTD

Proline residues in a protein can significantly change its conformational state. Proline-dependent conformational rearrangements of the 3D protein structure can lead to changes in the accessibility of epitopes. In this regard, it can be assumed that the proline residue is not necessarily included in the composition of the epitope. The use of site-directed mutagenesis enables determining, in detail, the location of epitopes on the surface of the HlyII C-terminal domain.

The interaction of mutant forms with HlyIIC-20 was studied using the HlyIIC-40 antibody, chosen from a panel of monoclonal antibodies against HlyIICTD obtained by the authors, as a control. Antibodies HlyIIC-20 and HlyIIC-40 both interacted with as the HlyIICTD well as full-length toxin in by the EIA and immunoblotting analysis. They were chosen because their K_aff_ values, which characterize their interaction with HlyIICTD, proved to be similar [15]. At the same time, they did not compete with each other to bind to the immobilized HlyIICTD during competitive EIA (see Figure 1). In this experiment, the antibodies (HlyIIC-20 and HlyIIC-40bio) were simultaneously added to the adsorbed HlyIICTD, which interacted with HlyIICTD and competed with each other. Therefore, these antibodies recognize non-overlapping regions on the surface of the HlyIICTD 3D structure.

As shown in Figure 2, HlyIIC-20 and HlyIIC-40 had different effects on the efficiency of the hemolysis of rabbit erythrocytes caused by hemolysins of strains ATCC 14579^T^, B771, and ATCC 4342^T^. Unlike HlyIIC-20, the monoclonal antibody HlyIIC-40 suppressed the hemolysis caused by HlyII of *B. cereus* ATCC 14579^T^ only partially and did not suppress the hemolysis caused by hemolysins of *B. cereus* strains, B771 and ATCC 4342^T^. These results also confirm that HlyIIC-20 and HlyIIC-40 interact with different epitopes on the surface of the HlyIICTD 3D structure.

### 2.2. Efficiency of HlyIIC-20 Binding to Full-Length Hemolysins of B. cereus Strains ATCC 14579^T^, B771, and ATCC 4342^T^

The work of [15] has demonstrated the influence of HlyIIC-20 on the hemolysis of rabbit erythrocytes caused by HlyII of *B. cereus* strains ATCC 14579^T^, B771 and ATCC 4342^T^, as well as by the mutant forms L324P (ATCC 14579^T^) and P324L (B771). It has been shown that the presence of the amino acid L324 in the primary sequence of HlyII is essential for the suppression of hemolysis. In turn, the presence of proline at position 324 in hemolysins reduced the level of protection against hemolysis. Figure 3 shows the interaction of HlyIIC-20 with toxins of the studied strains and their mutant forms in EIA. The substitution of L324P in the hemolysin of strain ATCC 14579^T^ led to a marked decrease in the efficiency of the antibody binding, which correlates with the data on the suppression of erythrocyte hemolysis. Hemolysis caused by HlyII of *B. cereus* strain ATCC 14579^T^ was suppressed by the HlyIIC-20 antibodies the most effectively. In contrast, the substitution of P324L in the hemolysin of *B. cereus* strain B771 led to an improvement in its interaction with the antibodies, which is also consistent with the data on hemolysis suppression. These data suggest that a mutation at position 324 alters the affinity of the interaction of HlyIIC-20 with HlyII.

In this case, however, HlyIIC-20 interacted most effectively with toxins of the *B. cereus* strain ATCC 4342^T^, but the hemolysis suppression of this strain via its antibodies was lower than for the ATCC 14579^T^ strain. Strain ATCC 4342^T^ features the presence of Leu at position 324.

The control antibody HlyIIC-40 showed the same trends when interacting with full-length toxins and their mutant forms. The principal difference was in the interaction with the full-sized HlyII of ATCC 4342^T^. The availability of the epitope for HlyIIC-40 did not practically differ, both for the free HlyIICTD and in the full-length toxin. It is possible that the accessibility of the epitope is also affected by differences in the amino acid sequences of the C-terminal domains of the hemolysins in these strains.

### 2.3. Determination of the HlyIIC-20 Epitope

To clarify the location of the HlyIIC-20 epitope on the surface of HlyIICTD based on the 3D structure (6d5z) determined via NMR [12], the results of the phage display described in [15] were taken into account. An analysis of the peptide amino acid sequences identified via phage display did not reveal identical motifs in the primary structure of HlyIICTD. This indicates that this epitope is conformational, i.e., the amino acid sequences of peptides obtained via phage display must correlate with the location of surface amino acid residues in the 3D structure of HlyIICTD. Analysis of the composition of the resulting peptides [15] showed that the formation of an epitope requires amino acid residues with amino groups (N, Q, or H) and a positively charged amino acid lysine or arginine (K, R). The amino acid sequence of the C-terminal domain does not contain histidines (H) or arginines (R). Therefore, this conformational epitope may include amino acid residues (N, Q, and K). Taking these data into account, mutant forms of HlyIICTD were created and the binding of HlyIIC-20 to these proteins was studied using EIA. Not only single but also multiple substitutions were created (two, three, or four amino acid residues were changed at the same time), since a single substitution may have had a minor effect on the interaction with the antibody. Multiple substitutions included amino acid residues located close together on the surface of HlyIICTD. When designing mutations, the authors decided to use the classic option of replacing amino acids with alanines [16]. After examining most of the mutant proteins, it turned out that all of the mutations affect both HlyIIC-20 and HlyIIC-40. It is possible that when replacing with alanine, hydrophobic clusters appear, although not large ones. In order to exclude the possibility that conformational changes in the protein depend on the type of substitution itself, some amino acid residues were replaced with glycine. Glycine substitutions should presumably not create additional hydrophobic sites on the surface of the protein and, in addition, may relieve steric tension in the protein structure that may arise due to the absence of a charged or hydrophilic amino acid that has been replaced. The P405M replacement was designed because this mutant form was studied and described in [14]. In this paper, all of the created mutations were confirmed via nucleotide sequencing. This analysis revealed occasional substitutions of N329S and N398D. These mutant forms were also tested for interactions with HlyIIC-20 and HlyIIC-40. Adding a positive charge by introducing aspartate to a triple N350A N352A Q353A mutant significantly reduced antibody binding.

Figure 4 shows the 3D structure of HlyIICTD and the position of the substituted amino acids on its surface. It was logical to assume that the substitution of amino acid residues included in the epitope recognized by HlyIIC-20 would affect its interaction with the mutant forms, and, in turn, the substitution of amino acid residues that are not included in the epitope should not affect this interaction. However, all the engineered substitutions of amino acid residues on the surface of HlyIICTD reduced the level of this interaction with varying degrees of efficiency. Each of the mutations localized in the investigated area of the surface led to a decrease in the binding of the mutant form to the HlyIIC-20 antibody (Figure 5). Such an effect of mutations on the binding of mutant forms to an antibody is possible under the assumption that amino acid substitutions affect the 3D structure of HlyIICTD, i.e., change the conformation of this protein. When HlyIICTD passes to another isoform, the mutual arrangement of amino acids on its surface changes, and the epitope is partially or completely “disrupted”, which leads to a decrease in the binding of the HlyIIC-20 to HlyIICTD.

This assumption was confirmed by the amino acid substitutions having a similar effect on the interaction of HlyIIC-40 with HlyIICTD. Figure 5 shows the results of the HlyIIC-40 binding to HlyIICTD in comparison with those for HlyIIC-20. Most amino acid substitutions designed affected the affinity of HlyIIC-40 for the mutant forms in the same way as HlyIIC-20. Herewith, some amino acid substitutions had a more significant effect on the level of antibody binding to HlyIICTD. For example, the addition of the N398D mutation to a mutant containing the N350A, N352A, or Q353A substitutions resulted in a significant reduction in the interaction.

HlyIIC-20 and HlyIIC-40 do not compete for binding sites with HlyIICTD. The identical effects of the same amino acid substitutions on the binding of completely different mAbs can only be caused by structural rearrangements in the HlyIICTD, which in turn affect the “structure” or accessibility of epitopes on the surface of the protein globule. At the same time, the double substitution of N339A and K340A did not affect the binding to HlyIIC-40 at all, but noticeably inhibited the binding to HlyIIC-20. A similar result was obtained for these mutations in combination with the P405M substitution. These data suggest that the substitution of N339A and K340A does not lead to the transition of HlyIICTD to another isoform, but affects the epitope for HlyIIC-20. Thus, the results of the comparative analysis indicate that the HlyIIC-20 epitope includes amino acid residues N339A and K340A.

Some amino acid substitutions show a more effective suppression of HlyIIC-20 recognition than HlyIIC-40. These amino acid substitutions—marked in Figure 4 as double N360A, N392A and triple Y367G, E368A, D369A—affect the recognition efficiency of HlyIIC-20 to a greater extent than they do the recognition efficiency of HlyIIC-40. It is possible that some of these amino acid residues, and, albeit less likely, all of these amino acids, are part of the HlyIIC-20 epitope.

The results obtained using site-directed mutagenesis confirm that the HlyIIC-20 and HlyIIC-40 epitopes are located in non-overlapping regions of the HlyIICTD protein globule.

## 3. Discussion

The presence of a proline residue in the polypeptide chain causes its bending; therefore, the presence or absence of this amino acid residue significantly changes the spatial structure of the protein, which in turn affects the accessibility or change of conformational epitopes. Substitutions of the proline residues at positions 324 and 405 resulted in significant changes in the affinity of HlyIIC-20 and HlyIIC-40 antibodies for the C-terminal domain (Figure 3 and Figure 5). The P405M mutation was chosen because the effect of P405 on the 3D structure of HlyIICTD was described in [14]. The ability of P405 to undergo cis/trans isomerization results in the existence of C-terminal domain isoforms. Replacing a tyrosine residue also has a significant effect on the 3D structure of the HlyIICTD (Figure 5). Herewith, almost every amino acid substitution, not only P405, in the protein sequence led to a change in the accessibility of epitopes recognized by HlyIIC-20 and HlyIIC-40, despite the fact that the introduced mutations were localized in areas both close to and remote from each other on the surface of the protein globule. These results indicate the high lability of the HlyIICTD 3D structure.

It has been shown that pre-treatment of HlyII with HlyIIC-20 prevents hemolysis by inhibiting the oligomerization stage [15]. The data obtained enable an assumption of the manner in which HlyIIC-20, unlike other antibodies, prevents hemolysis caused by HlyII of the ATCC 14579^T^ strain. We hypothesize that HlyIIC-20 stabilizes the structure of HlyIICTD, decreasing its lability and thereby preventing its transition into different isoforms. The lability of HlyIICTD in turn affects the assembly of the pore-forming oligomer HlyII, as the result of which hemolysis is prevented [15].

The stabilizing effect of HlyIIC-20 on HlyIICTD is confirmed by the data in Figure 1, which show that the interaction of immobilized HlyIICTD with HlyIIC-20 leads to a statistically significant and reliable increasing (of up to 18%) in HlyIIC-40 binding with HlyIICTD. Apparently, upon the binding of HlyIIC-20 to HlyIICTD, the 3D structure of HlyIICTD is stabilized, which can lead to an increase in the efficiency of the binding of HlyIIC-40 to HlyIICTD. The interaction of a full-size toxin with the membranes of a target cell proceeds through a stage of monomer oligomerization, followed by the formation of a pre-pore and then of a mature pore capable of passing ions and other low-molecular substances, which leads to cell death. Each of these stages is accompanied by conformational rearrangements [11]. Thus, the lability of the 3D structure of HlyIICTD corresponds to the general concept of its pore formation in the presence of the target cell membrane.

The identification of epitopes is important in the development of highly specific and harmless vaccines, and therapeutic and diagnostic antibodies [17,18], since epitopes, namely, allow the immune system to recognize and respond to specific pathogens [19]. Therefore, it is especially important to determine the epitope of an antibody capable of neutralizing one of the significant pathogenic factors of *B. cereus* in order to design a mimetic, based on which a protection against the action of HlyII of various *B. cereus* strains can be created.

HlyII of *B. cereus* is a member of the β-barrel pore-forming toxin family, the closest relative of which, regarding its amino acid composition, is the well-studied α-hemolysin of *Staphylococcus aureus* [20,21]. In this regard, the *S. aureus* α-hemolysin model [22] is used to analyze the putative 3D structure of the full-length HlyII of *B. cereus*. The structure of the toxins in this group is characterized by the presence of a large number of β-sheets and is formed predominantly of amino acid residues from different parts of the polypeptide chain linked together; that is to say, epitopes on the surface of a molecule are most likely conformational, representing a 3D structure formed as a result of protein folding.

The conformational epitope of HlyIIC-20 includes amino acid residues N339A and K340A, located one after the other. In this work, the substitution of extra-epitope amino acid residues has been shown to alter the accessibility of conformational epitopes of HlyIICTD. In our opinion, one of the most interesting results of this work is that it shows the lability of the HlyIICTD 3D structure. This result explains the difficulties that arise when searching for antibodies that inhibit the functioning of mobile proteins, in particular toxins, the structure of which changes greatly during their activity.

## 4. Materials and Methods

### 4.1. Strains, Plasmids and Enzymes

Restriction endonucleases KpnI and NdeI (Thermo Scientific, Waltham, MA, USA), T4-DNA ligase (NEB, Ipswich, MA, USA), protein markers and DNA electrophoresis markers (Thermo Scientific, Vilnius, Lithuania), TaqSE-DNA polymerase (SibEnzyme, Moscow, Russia), and dNTP mix (Thermo Scientific, Waltham, MA, USA) were used. The PCR product was cloned using the pET29b (+) vector NdeI and KpnI.

### 4.2. Molecular Cloning and Site-Directed Mutagenesis

The pET29-containing HlyIICTD sequence was used as a template for the subsequent mutagenesis experiments. The HlyIICTD mutants were generated using an overlap extension PCR [23]. The HlyIICTD fragments were amplified using the overlap primers shown in Table 1. For all mutants, the overlapping region of the primers contained the mutation of interest. All plasmids had been previously verified via sequencing before transformation into the expression strain BL21(DE3).

### 4.3. Expression and Purification of HlyIICTD His6 and Its Mutant Forms

Expression and purification of HlyIICTD His6 and its mutant forms was carried out as described in [11].

### 4.4. Enzyme immunosorbent Assay

HlyIICTD, HlyII, and their mutant forms were adsorbed on the surface of the wells of EIA-immune plates (Sovtech, M-011, Novosibirsk, Russia) from solution (1 µg/mL^−^^1^, 50 µL) overnight at 4 °C. The free binding sites of the experimental wells were blocked with PBST (PBS containing 0.1% Tween 20) for 30 min. Then, the antibody solution in PBST was introduced into the plate wells. Incubation with the antigen was carried out for 1 h at 37 °C. Then, the plates were washed with PBST no less than 6 times, and a conjugate of antibodies against mouse immunoglobulins was added (Thermo Scientific 31432, Waltham, MA, USA, Goat anti-Mouse IgG [H+L] HRP conjugate) in PBST, diluted as indicated by the manufacturer and incubated at 37 °C for 1 h. A 4 mm solution of ortho-phenylenediamine peroxidase substrate in a citrate–phosphate buffer (26 mM citric acid, 50 mM Na_2_HPO_4_, pH 5.0) containing 0.003% (*v*/*v*) H_2_O_2_ was used for detection. After it developed a color, the reaction was stopped by adding an equal volume of 10% (*v*/*v*) sulfuric acid, and optical absorption was measured at 490 nm using an iMark microplate reader (Bio-Rad, Irvine, CA, USA).

In the case of the inhibition of the interaction of biotinylated HlyIIC-40 with immobilized HlyIICTD via unlabeled HlyIIC-20, solutions composed of biotin-labeled and unlabeled antibodies were simultaneously introduced into the wells of EIA-immune plates in PBST. The wells were pretreated sequentially with HlyIICTD and PBST. The reaction was carried out as described in the previous paragraph. Biotinylated antibodies bound to HlyIICTD were detected using horseradish peroxidase-conjugated streptavidin (Thermo Scientific, 21126, MA, USA) according to the manufacturer’s instructions.

### 4.5. Conjugation of Antibodies with Biotin

A solution of antibodies (1 µg/mL^−^^1^) in 0.1 M bicarbonate buffer (pH 9.0) was supplemented with biotin N-hydroxysuccinimide ester (Sigma-Aldrich, H1759, St. Louis, MO, USA) in dimethylsulfoxide (Sigma-Aldrich, 471267, MO, USA) (1 µg/mL^−^^1^) at a molar ratio of 1:20. After incubation for 4 h at room temperature, the solution was dialyzed against PBS.

### 4.6. Measurement of Hemolytic Activity of Hemolysin II in the Presence of mAbs

In a 96-well microplate, a series of 2-fold decreasing dilutions of hemolysin II (from 32 HU/mL) were incubated with an equal volume of monoclonal antibodies at a constant concentration of 0.5 μM for 15 min at 37 °C. An equal volume of 1% rabbit erythrocyte suspension was added to the mixtures (HlyII + mAbs). The mixtures were incubated for 30 min at 37 °C. The hemolytic activity was measured as described in [24].

## Figures and Tables

**Figure 1 ijms-24-16437-f001:**
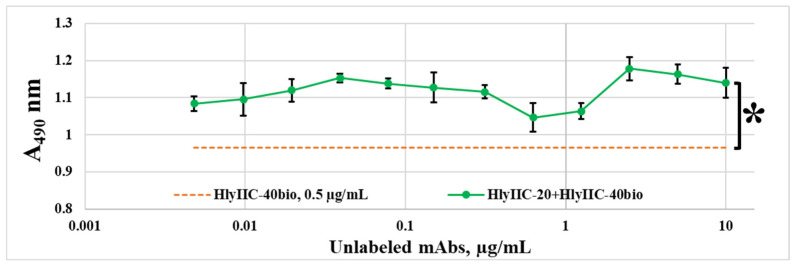
Inhibition (in the EIA) of the interaction of immobilized HlyIICTD with HlyIIC-40bio via unlabeled HlyIIC-20. The reaction was visualized with streptavidin conjugated with peroxidase. Data are presented as the mean ± standard error of the mean values for three independent the tests; * shows statistically significant differences (*p* < 0.05, Mann–Whitney test) between the study groups marked with parentheses.

**Figure 2 ijms-24-16437-f002:**
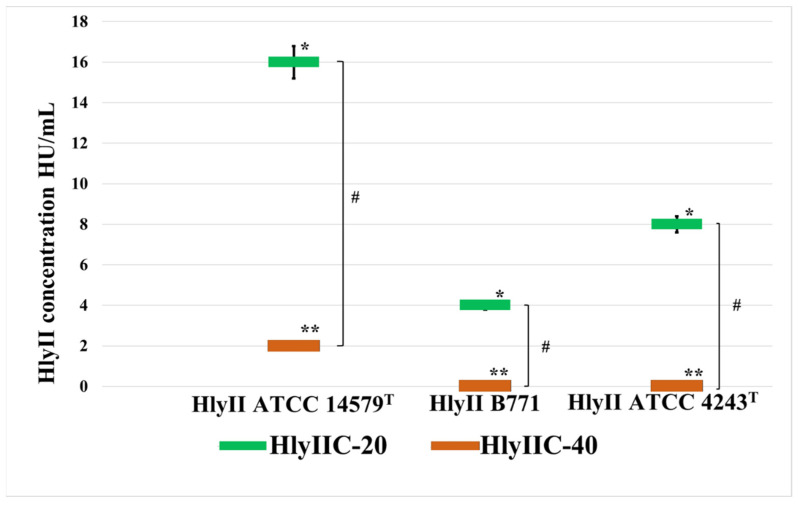
The influence of mAbs HlyIIC-20 and HlyIIC-40 on the hemolytic activity HlyII of various *B. cereus* strains. Threshold concentrations of the hemolysins are shown at which 0.5 μM mAbs completely inhibits hemolysis. *—statistically significant differences (*p* < 0.05, Mann–Whitney test) for HlyIIC-20 when comparing its effect on HlyII of different strains; **—statistically significant differences (*p* < 0.05, Mann–Whitney test) for HlyIIC-40 when comparing its effect on HlyII of different strains; #—statistically significant differences (*p* < 0.05, Mann–Whitney test) when comparing the effect of HlyIIC-20 and HlyIIC-40 antibodies on HlyII of the same strain.

**Figure 3 ijms-24-16437-f003:**
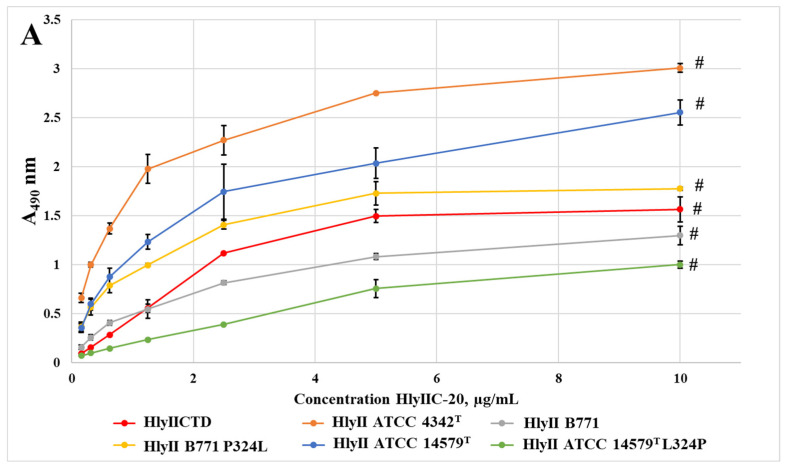
Interaction of mAbs HlyIIC-20 (**A**) and HlyIIC-40 (**B**) with immobilized HlyIICTD, full-sized toxins, and mutant forms of toxins. Data are represented as the mean ± SEM values of three independent of the tests; # shows statistically significant differences (*p* < 0.05, Mann–Whitney test).

**Figure 4 ijms-24-16437-f004:**
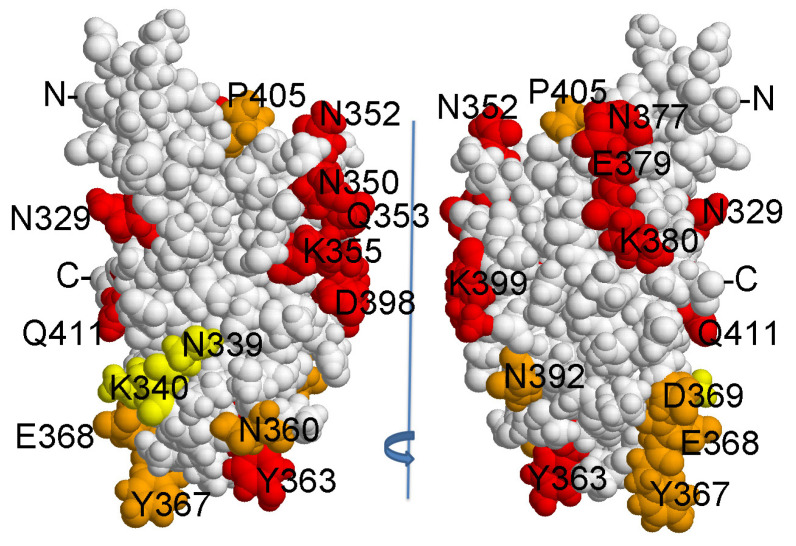
Location of amino acid substitutions on the surface of HlyIICTD (6d5z). Orange and yellow colors indicate statistically significant differences in the binding of HlyIIC-20 with HlyIICTD from the values for HlyIIC-40. Yellow—amino acid residues that are part of the HlyIIC-20 epitope; red—the remaining substitutions of amino acid residues.

**Figure 5 ijms-24-16437-f005:**
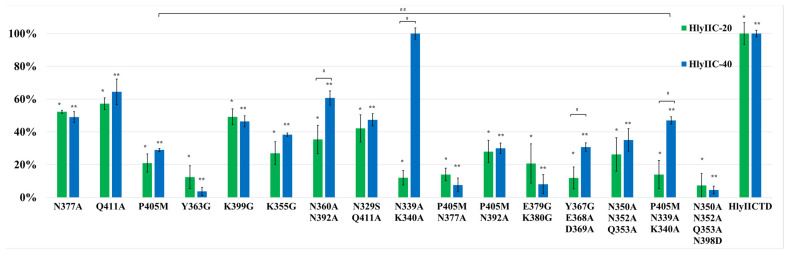
Comparison of the interaction of HlyIIC-20 and HlyIIC-40 with immobilized HlyIICTD and its mutant forms in ELISA. #—statistically significant differences (*p* < 0.05, Mann–Whitney test) between the studied antibodies to various mutant forms of HlyIICTD; ##—statistically significant differences (*p* < 0.05, Mann–Whitney test) between the binding of HlyIIC-40 with mutant forms P405M and P405M N339A K340A; *—statistically significant differences (*p* < 0.05, Mann–Whitney test) in the binding of HlyIIC-20 to mutant forms compared to its binding to HlyIICTD; **—statistically significant differences (*p* < 0.05, Mann–Whitney test) in the binding of HlyIIC-40 to mutant forms compared to its binding to HlyIICTD.

**Table 1 ijms-24-16437-t001:** Primers used for HlyIICTD site-directed mutagenesis.

Mutation Name	Oligonucleotide Name	Sequence 5′->3′
N377A	F_N377A	CTTTGTAGCTGGTGAAAAGGTCTATAC
	R_N377A	TTCACCAGCTACAAAGATACCCCAAT
Q411A	R_Q411A	TAACTCGAGGGTACCGATAGCTTTAATCTCGATATAAGGTCC
P405M	F_P405M	AAAGGAATGTATATCGAGATTAAACAGATC
	R_P405M	GATATACATTCCTTTAATGTTTAATTTG
Y363G	F_Y363G	GCTGGTGGAGGTATCAGTTACGAAG
	R_Y363G	GATACCTCCACCAGCATTGCTAGATG
K399G	F_K399G	GATATTAACGGATTAAACATTAAAGGACCTTATATC
	R_K399G	AATGTTTAATCCGTTAATATCATTAGAGATATTGC
K355G	F_K355G	CAACTTGGAGCTACATCTAGCAATGC
	R_K355G	GATGTAGCTCCAAGTTGATTTCCATTC
N360A N392A	F_N360A	TCTAGCGCAGCTGGTTATGGTATC
	R_N360A	ACCAGCTGCGCTAGATGTAGCTTTAAG
	F_N392A	GTAGGCGCTATCTCTAATGATATTAACAAAT
	R_N392A	AGAGATAGCGCCTACAGTTGATTTTTC
N339A K340A	F_NK339	AACTTGCTGCTGGAAAAGGGAAATTATC
	R_NK339	CTTTTCCAGCAGCAAGTTTATCATTCACGC
P405M N377A	F_P405M	AAAGGAATGTATATCGAGATTAAACAGATC
	R_P405M	GATATACATTCCTTTAATGTTTAATTTG
	F_N377A	CTTTGTAGCTGGTGAAAAGGTCTATAC
	R_N377A	TTCACCAGCTACAAAGATACCCCAAT
P405M N392A	F_P405M	AAAGGAATGTATATCGAGATTAAACAGATC
	R_P405M	GATATACATTCCTTTAATGTTTAATTTG
	F_N392A	GTAGGCGCTATCTCTAATGATATTAACAAAT
	R_N392A	AGAGATAGCGCCTACAGTTGATTTTTC
E379G K380G	F_EK379G	GGTGGTGGAGTCTATACTTTTAATGAAAAATCAAC
	R_EK379G	GTATAGACTCCACCACCATTTACAAAGATACCC
Y367G E368A D369A	F_YED367GAA	CAGTGGAGCAGCAAAAAATTGGGGTATCTTTG
	R_YED367GAA	TTTTGCTGCTCCACTGATACCATAACCAG
P405M N339A K340A	F_P405M	AAAGGAATGTATATCGAGATTAAACAGATC
	R_P405M	GATATACATTCCTTTAATGTTTAATTTG
	F_NK339	AACTTGCTGCTGGAAAAGGGAAATTATC
	R_NK339	CTTTTCCAGCAGCAAGTTTATCATTCACGC
N350A N352A Q353A	F-NNQ	ATGGCTGGAGCTGCTCTTAAAGCTACATCTAGC
	R-NNQ	AAGAGCAGCTCCAGCCATTGAAAGAGATAATTTC

## Data Availability

Data is contained within the article.

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
