# Peer review of "Utilizing Extraepitopic Amino Acid Substitutions to Define Changes in the Accessibility of Conformational Epitopes of the Bacillus cereus HlyII C-Terminal Domain"

_ijms, 2023, doi:10.3390/ijms242216437_

Round 1
Reviewer 1 Report
Comments and Suggestions for Authors
The overall of this paper is fine, I have a few of concerns as in attachment.

some sentence need to be modified.
Author Response
The authors are grateful to the reviewer for high opinion, carefully reading the manuscript and for comments.
- Author emphasizes the importance of proline residues, but I did not get the point. In Figure 5, the efficiency of P405M is not significant as Y363G mutation.
The P405M mutation was chosen because the effect of P405 on the 3D structure of HlyIICTD was described in [14]. The ability of P405 to undergo cis/trans isomerization results in the existence of C-terminal domain isoforms. In this work, several mutations were made with Pro 405, each of which led to a noticeable decrease in binding to antibodies, so the authors, in the case of replacement of proline residues, allowed a generalization about the effect of this amino acid residue on the conformation of the protein under study. We agree that substitution of a tyrosine residue also has a significant effect on the 3D structure of the protein. The corresponding addition to the text has been made.
- Some sentences need to be modified, such as in line 69, “in [15] we have described ….” I think it should be changed to “ in previous report [15] we have dexribed…”; in line 79, “Proline amino acid residues…” should remove amino acid or residues, these two words are repeated.
Corrections to the text of the manuscript were made.
- Figure1, in order to make the figure looks better, in my opinion it is better to set up the X Axis scale from 0.8-1.4
The figure has been modified.
- Why some amino acid substitute to Alanine, and some substitute to Glycine (Y363G), some mutate to aspartic acid? In order to compare their function, they all should be mutated to Alanine.
The authors added to the text a more detailed presentation of the logic for choosing amino acid substitutions in HlyIICTD, lines 166-184.
- Figure 5, it’s better to reorganize the order of the mutation data, Data are shown in order of single, double, and triple mutations, as well as with increasing the number of residues.
The figure has been modified.

Reviewer 2 Report
Comments and Suggestions for Authors
It is well and concisely written paper with relevant work focussed on the aim. This work has significance, as this work entails, and addresses a disease related target and hence helps in understanding, improving, designing and testing therapeutic modalities.
The work also reflects the importance of few amino acids in protein conformation and epitope residue authenticity, especially in non linear or conformational epitope targets, and adds further understanding of the antibody and antigen interaction or binding, especially inhibiting or neutralizing antibodies to a potential disease causing agents.
Authors mention about phage display, where you say, "taking into account the results of phage display", and it would be helpful to the reader if you could elaborate or summarize or refer to the findings, backing this statement.
I am curious if antibody mapping (both antibodies HlyIIC-20 and/or HlyIIC-40), using the phage peptide libraries. In my experience, phage peptide libraries can yield peptide phage with conformational epitopes, often with critical amino acids of the original epitope. If not for this current work, you could comment on this.
Overall, this is important contribution in understanding the disease related antigen's sequence, structure and function.
Author Response
The authors are grateful to the reviewer for high opinion, carefully reading the manuscript and for comments.
Authors mention about phage display, where you say, "taking into account the results of phage display", and it would be helpful to the reader if you could elaborate or summarize or refer to the findings, backing this statement.
I am curious if antibody mapping (both antibodies HlyIIC-20 and/or HlyIIC-40), using the phage peptide libraries. In my experience, phage peptide libraries can yield peptide phage with conformational epitopes, often with critical amino acids of the original epitope. If not for this current work, you could comment on this.
The authors expanded the description of the phage display results, lines 156-165. We believe that with a successful combination of circumstances, it is theoretically possible for phage display to determine a conformational epitope. In this case, phage display helped to identify only key amino acid residues on the surface of the protein globule under study.

Reviewer 3 Report
Comments and Suggestions for Authors
Comments to authors
The manuscript by Rudenko et al is a follow-up study of a previously published article from the same group. The authors aimed to investigate and identify the epitopes of a monoclonal antibody (HlyIIC-20) by using the mutagenesis approach. The authors study the impact of several mutations in the on the surface of HlyIICTD protein and assess their effect on binding of the HlyIIC-20 mAb in comparison to another mAb HlyIIC-40. Using the panel of mutants the authors identify the key residues that may serve as key epitopes for the HlyIIC-20 mAb and that the epitopes of both Abs are non-overlapping. The study is generally well and thoroughly done. The article is well written, presented and supported by appropriate statistical analysis on all the figures.
Line 50: The authors write that “ Elimination of the C-terminal domain reduces the hemolytic activity towards rabbit erythrocytes eightfold”. Not a comment but for my own curiosity, is there a similar reduction in hemolytic activity towards human erythrocytes?
Line 53: “At a prolonged exposure to HlyIICTD, it lyses monocytes, macrophages, and T cells [11].” The authors should modify this sentence to specify that these are primary human cells or representative human cell lines.
What do the authors think of the biphasic reaction in Fig 1?
Suppression of hemolysis by L324P and P324L is referenced from an earlier study (page 4, line 121-127). Perhaps include the main findings from the article referenced.
Line 174: “…mutant containing the substitutions N350A, N352A, or Q353A…” Should be and not or as it shows the effect on a quadruple mutant. The authors should include reasoning why the N398D mutant was not studied individually and the basis of selecting the positions to be included in the double/triple/quadruple mutants.
Figure 5 needs explanation why some of the sites are not shown as single mutation but those amino acids are included in double or triple mutants. Also same position is changed to different AA as single mutant (K399G) while double mutant compared changes into a different amino acid (N339A, K340A) making it hard to really compare the binding.
Line 255: “Substitution of extra-epitope amino acid residues has 253 been shown to alter the accessibility of conformational epitopes of the C-terminal domain 254 of B. cereus HlyII.” Include the citation.
Figure 5: The single mutant P405M decrease the binding of both mAb similarly, but the triple mutant (P405M, N339A K340A) increases the binding of HlyIIC-40 mAb vs P405M. Is that statistically significant?

Author Response
The authors are grateful to the reviewer for high opinion, carefully reading the manuscript and for comments.
Line 50: The authors write that “Elimination of the C-terminal domain reduces the hemolytic activity towards rabbit erythrocytes eightfold”. Not a comment but for my own curiosity, is there a similar reduction in hemolytic activity towards human erythrocytes?
The authors did not find any literary data on a similar reduction in hemolytic activity towards human erythrocytes and erythrocytes of other species.
Line 53: “At a prolonged exposure to HlyIICTD, it lyses monocytes, macrophages, and T cells [11].” The authors should modify this sentence to specify that these are primary human cells or representative human cell lines.
Clarification included in the text.
What do the authors think of the biphasic reaction in Fig 1?
In the experiment, the results of which are shown in Fig. 1, antibodies (HlyIIC-20 and HlyIIC-40 bio) were simultaneously added to the sorbed HlyIICTD, which interacted with the antigen, competing with each other. The reaction took place in one stage.
Suppression of hemolysis by L324P and P324L is referenced from an earlier study (page 4, line 121-127). Perhaps include the main findings from the article referenced.
Key findings on hemolysis suppression by L324P and P324L added to manuscript, lines 124-127.
Line 174: “…mutant containing the substitutions N350A, N352A, or Q353A…” Should be and not or as it shows the effect on a quadruple mutant. The authors should include reasoning why the N398D mutant was not studied individually and the basis of selecting the positions to be included in the double/triple/quadruple mutants.
The N398D mutant was not studied individually, as it was not planned and was discovered during nucleotide sequencing. Mutant forms containing more than one substitution were designed based on the proximity of the replaced amino acid residues on the surface of HlyIICTD. All necessary explanations are included in the text, lines 166-184.
Figure 5 needs explanation why some of the sites are not shown as single mutation but those amino acids are included in double or triple mutants. Also same position is changed to different AA as single mutant (K399G) while double mutant compared changes into a different amino acid (N339A, K340A) making it hard to really compare the binding.
The authors added to the text a more detailed presentation of the logic for choosing amino acid substitutions in HlyIICTD, lines 166-184.
Line 255: “Substitution of extra-epitope amino acid residues has been shown to alter the accessibility of conformational epitopes of the C-terminal domain of B. cereus HlyII.” Include the citation.
Corrected.
Figure 5: The single mutant P405M decrease the binding of both mAb similarly, but the triple mutant (P405M, N339A K340A) increases the binding of HlyIIC-40 mAb vs P405M. Is that statistically significant?
Comparison of the interaction of HlyIIC-40 with mutant forms of HlyIICTD P405M and P405M N339A K340A also demonstrates statistically significant differences (P < 0.05, Mann-Whitney test). Statistical analysis added to the figure 5.
